# Color Stability of Resin Cements after Water Aging

**DOI:** 10.3390/polym15030655

**Published:** 2023-01-27

**Authors:** Claudia Mazzitelli, Gaetano Paolone, Joseph Sabbagh, Nicola Scotti, Alessandro Vichi

**Affiliations:** 1Department of Biomedical and Neuromotor Sciences DIBINEM, University of Bologna, 40125 Bologna, Italy; 2Department of Dentistry, IRCCS, San Raffaele Hospital, Dental School, Vita Salute University, 20158 Milan, Italy; 3Department of Restorative Dentistry and Endodontics, Lebanese University, Beirut 1533, Lebanon; 4Department of Surgical Sciences, Dental School Lingotto, University of Turin, 10126 Turin, Italy; 5Dental Academy, University of Portsmouth, William Beatty Building, Hampshire Terrace, Portsmouth PO1 2QG, UK

**Keywords:** resin cement, color stability, water aging, self-adhesive cement, dual-cure

## Abstract

The color stability of resin cements plays a key role in the achievement of esthetically-pleasant restorations. Resin luting materials can be mainly divided into two main classes: adhesive (relying on previous application of adhesive systems) or self-adhesive (also known as one-step cements). The different chemical compositions determine their physio-mechanical characteristics which, in turns, influence their color stability. To evaluate the color variations of different dual-cured resin cements after water aging, 80 disc-shaped specimens (15 mm in diameter and 1.2 mm thick) were obtained from the following resin cements (*n* = 10): (1) Maxcem Elite Universal, MCU (Kerr); (2) RelyX Universal, RXU (3M); (3) Calibra Ceram, CAL (Dentsply); (4) Multilink, MUL (Ivoclar-Vivadent); (5) Panavia V5, PAN (Kuraray); (6) Calibra Universal, CUN (Dentsply); (7) SpeedCEM Plus, SCP (Ivoclar); and (8) Panavia SA, PSA (Kuraray). After light-polymerization, the specimens were measured with a spectrophotometer and CIELab* values were recorded. The specimens were then placed in a digitally controlled thermostatic water bath at 60° for 30 days and afterwards the color measurements were repeated. Color differences were calculated for each specimen before and after water-aging procedures with ΔEab formula and the data were statistically analyzed (*p* < 0.05). The type of cement statistically influenced the ΔEab (*p* < 0.05), with MCU showing the lowest color variations (4.3 ± 0.7) whereas RXU and PSA the highest (16.9 ± 1.6 and 16.8 ± 1.2, respectively). No differences were observed between CAL, CUN and SCP (*p* = 0.05). Color stability is related to the chemical composition of the resinous luting materials, thus material dependent.

## 1. Introduction

The advancement in materials technology has remodeled the restorative/prosthetic dental treatments. The availability of increasingly performing restorative materials, along with the enhancement of adhesive cementation, has made possible the realization of minimally invasive restorations with high aesthetic standards, reinforced mechanical performances, and longer prognosis over time [1,2].

Several factors may influence the longevity of an indirect restoration [3,4]. Among them, the luting material has a relevant role. The rationale for the selection of the luting material is essential to confer the restoration reliable strength and optimal esthetic [2]. In general, the cement is a physical ligand between the restoration and the tooth, providing retention at both interfaces to withstand occlusal chewing forces, and marginal sealing to prevent secondary caries. Among the several materials available for cementation of indirect restorations, resin-based cements represent the materials of choice, thanks to their physio-mechanical characteristics which allow to perform treatments even in non-retentive clinical situations, such as when preparations are very limited [5]. In brief, resin cements are mostly divided into two main groups, according to the previous need of tooth/restoration conditioning: adhesives (relying on previous application of adhesive systems) and self-adhesives (also referred as one-step cements). Recent advancement in material’s technology has moved to simplify and create less-technique sensitive luting materials. In this scenario, the introduction of self-adhesive resin cements in 2004 has initiated an interesting pathway through the refinement of formulations and techniques leading to the latest marketed universal resin cements. Decreased operator’s mismanipulation and reduced operative time is attracting the interest of marketing, clinicians, and researchers who individuate simplified self-adhesive cements as the materials of the future.

Irrespective of the class to which they belong, resin cements are available in different shades that can be chosen by the clinician depending on the clinical situation [6], the material used to fabricate the restoration—ceramic, zirconia or composite [7,8] —and the translucency and thickness of the restoration [8]. The proper selection of the color of the cement is particularly meaningful when dealing with esthetically demanding restorations, and clinicians must be aware that a wrong choice of cement color can affect the final aesthetic result of the restoration [9,10].

Resin cements are methacrylate-based materials which differ in terms of filler content and photoinitiators, concurring in defining their mechanical behaviors and clinical performances [11,12]. In general, a chemical and hydrolytic instability commonly characterize resin cements [1]. This is particularly evident in those situations where the margins of the restorations are located close to the gingival sulcus, possibly exposing the cement to the crevicular fluids and oral environment [13]. The tendency of resin-based materials to absorb water is one of the phenomena that mostly causes chemical and mechanical degradation in terms of polymer swelling, mechanical deterioration, debonding and fracture, as well as material discoloration [1,13,14,15,16,17].

The entity of color variation may be related to the chemical composition of the material itself in terms of filler content or activation mode [14,15,16,17,18], or dependent on external factors, e.g., adsorption media or aging [17,19]. Regardless of the cause of discoloration, the chromatic instability of the cement has an impact on the final shade of the restoration which may require early replacement to reestablish esthetics, possibly causing patient’s dissatisfaction and impacting on the economical aspect.

Given the importance that chromatic variability of resinous cements has on the final shade of the restorations and its influence on the esthetic outcome, and taking into account the chemical variabilities existing among the different resin-based luting materials currently on the market, the present study aimed at evaluating the color changes of different dual-cured resinous cements (either multi-step adhesive and one-step self-adhesive luting agents) when submitted to accelerated water aging. In particular, the null hypotheses tested were that: (i) color stability is not affected by water aging; and that (ii) no differences in color stability exist between the resin cements tested.

## 2. Materials and Methods

Six self-adhesive resin-based cements with different chemical compositions and manufacturers (Calibra Ceram, Dentsply Sirona; Multilink N, Ivoclar; Panavia V5, Kuraray Noritake; Calibra Universal, Dentsply Sirona; SpeedCEM Plus, Ivoclar), and two universal resin cements (Maxcem Elite Universal, Kerr; RelyX Universal, 3M) were used for the study. The most available translucent shade was selected for each material. Table 1 shows the details related to the tested resin cements.

For each material, 10 specimens were prepared according to the recommendations of ISO 4049:2019 [20]. A disc of steel with a through hole and marked with an identifying number was placed over a PVC sheet (dental sheets for cementation, Henry-Schein, NY, USA). The diameter of the hole was 15 mm, and the thickness of the steel disk was 1.2 mm.

The hole was filled with the cement with great attention to avoid major overfilling. Then another sheet of PVC was applied and slightly pressed with a microscope glass to avoid porosities or air entrapment within the material. The discs were light-cured for 10 s with a polymerization lamp (Kerr Demi Ultra, Kerr, Orange, CA, USA) with a power output of 1.100 (base)/1.330 (peak) mW/cm^2^ (Figure 1).

At the end of the polymerization process, glass and PVC sheets were carefully removed and the thickness of each composite disc was checked with a digital micrometer with an accuracy of ±0.01 mm. Specimens with thickness of 1.2 ± 0.05 mm were included in the study. Specimens were then left undisturbed at room temperature at dark for 24 h to allow for post polymerization.

A spectrophotometer (Ocean Optics PSD1000, Orlando, FL, USA) equipped with a 10.0 mm opening integrating sphere (Ocean Optics ISP-REF, Orlando, FL, USA) was used for color analysis. The spectrophotometer was connected to a computer running color measurement software (OOILab 1.0, Ocean Optics, Orlando, FL, USA) (Figure 2).

D65 illumination and a 10° standard observation angle were selected. The output of the spectrophotometer was set over 10 scans. As a background, an A3 Vitablocs Mark II I-40 CAD/CAM block (VITA Zahnfabrik, Bad Säckingen, Germany) was used (15.5 × 19 × 39 mm^3^). Values were recorded in CIELAB color coordinate system.

After performing color measurement, specimens were placed in a digitally controlled thermostatic water bath at 60 °C for 30 days [21,22]. Water was changed every 7.5 days. After the aging process, specimens were removed from the bath, cleaned in an ultrasonic bath for 3 min, cleaned and dried with oil-free air spray, and then the spectrophotometric measurements were performed with the same procedure as previously described.

To calculate the color changes (∆E), the following formula was used [23]:∆E = [(L*_1_ − L*_2_)^2^ + (a*_1_ − a*_2_)^2^ + (b*_1_ − b*_2_)^2^]^1/2^
with L* representing the brightness (from 0 = black up to 100 = white), a* was the redness/greenness (from -a = green up to +a = red), and b* was the yellowness/blueness (from −b = blue up to +b = yellow), all of them before (1) and after (2) water aging.

The ΔE values obtained were statistically analyzed using the software SigmaPlot for Windows version 11.00 (Systat Software, Inc., San Jose, CA, USA). As the data distribution was normal according to the Kolmogorov–Smirnov test (*p* > 0.05), and group variances were homogenous according to the Levene test (*p* > 0.05), a One-Way Analysis of Variance (ANOVA) was applied, followed by the Tukey test for post hoc comparisons. The level of significance was set at *p* = 0.05.

## 3. Results

The color differences (ΔE_ab_) obtained after water aging per each resin cement tested are presented in Table 2.

The One-Way ANOVA and Tukey test post hoc comparisons showed a statistically significant difference between groups (*p* < 0.001). Figure 3 graphically illustrates the mean in ΔE_ab_ among the tested materials.

Considering the adhesion mode, differences were found between universal and self-adhesive groups. In the first, MCU attained the highest color stability when compared to RXU (*p* < 0.05), and the latter showed the highest color changes along with PSA. In general, MCU demonstrated the lowest color differences thus the highest color stability after water aging among the tested groups (4.3 ± 0.7), followed by PAN (6.0 ± 0.7) with the difference between the two cements statistically significant (*p* < 0.05). Statistically significant differences were also found among the self-adhesive cements. No differences were observed between SCP (7.5 ± 0.8), CUN (7.8 ± 1.3) and CAL (7.9 ± 1.0), although statistically significant differences were found when comparing them to MCU and PAN. The higher color variations, statistically significant from all the other groups, were attained by PSA (14.1 ± 0.7) and RXU (16.9 ± 1.6), although they were statistically comparable between themselves (*p* = 0.05).

Concerning with color parameters L*, a*, and b* when independently evaluated, the behavior of the various materials was different. Regarding L* color coordinate, MUL showed the highest color change, followed by CUN and CAL. The a* color coordinate is typically not subject to significant changes in the dental color space, however, our study found that the color shift of MUL was noteworthy. Regarding the b* color coordinate, the change was more evident in RXU and PSA, whereas MAX showed the lowest change (Table 3).

## 4. Discussion

Based on the results of the present study, water aging influenced the color stability of the resin cements tested and differences were observed between materials. Accordingly, the two null hypothesis that color stability is not affected by water aging and that no differences in color stability exist between the resin cements tested have to be rejected.

The color stability of the cement is an important characteristic to consider during the decision making of the luting agent [24], as color mismatches of the cement can affect the final shade of the restoration, both for CAD/CAM lithium-disilicate crowns [25,26], as well as for heat-pressed lithium disilicate veneers [27,28], and felspathic ceramic restorations [29]. With such translucent prosthetic materials, the optical properties (translucency and color) of the resin cement can influence the final shade of metal-free restorations, with translucent cement having a lower efficacy on the color mismatch [30]. Particularly, when dealing with anterior restorations, color mismatch is one of the reasons leading to remaking of the restorative manufact with consequent patient’s dissatisfaction and increased costs.

Generally, color stability refers to the ability of a material to maintain the original color, or at least limit color changes in clinical conditions. Considering the high survival rate of porcelain laminate veneers [31] and anterior ceramic veneers [32], it is unavoidable to consider a certain exposure of the resin luting agents at the margin of the restorations cautiously exposing them to discoloration and impacting with the final esthetics [31].

Among the different types of luting materials, universal resin cements represent the latest category of luting materials introduced in the market, claimed to be used with/out the previous application of the bonding system according to the dentist’s choice and clinical situations [33]. From an esthetic point of view, the polymerization mode of the resin cement may influence their color stability, with light-cured materials that have been reported less color changes than dual or chemically cured materials [34,35,36]. The absence of tertiary amines in light-cured cements has been described to be the main cause for this favorable behavior [35,37,38]. On the other hand, self-cured or dual-cured cements have been found to be preferred when the light transmission is hindered by the presence of restorations of a certain thickness or by clinical and anatomical criticality [39,40,41,42].

Resin cements have a common structure (methacrylate monomers with small filler contents, redox initiators, and photoinitiators). This characteristic makes them all, in a material-dependent way, prone to chemical instability and hydrolytic degradation phenomena, due to the ability of resin monomers to absorb water [1,42,43]. Previous investigations have reported that the hydrolytic degradation can influence the mechanical properties of the material causing polymer swelling, degradation of the filler–matrix interface, weaking of the polymer network, and discoloration of the material itself [11,17,44]. The color changes observed after water absorption have been specifically linked to dual-cure resin cements that contain amine-based co-initiators, which are known to be easily affected by oxidation processes [12].

Water aging has been adopted to simulate a material’s degradation as hydrolytic phenomena used to occur in the resin matrices. The adsorption of water into the cement causes expansion and matrix swelling. The disruption at the resin matrix–inorganic filler level caused by water sorption can create microcracks formations [41,45] with consequent risk of discoloration of the cement [37]. This sequence of detrimental events has been ascribed to the chemical composition of the cements, and particularly to their organic matrices and polymerization modes [13,44,46,47]. Water sorption of RBCs is in fact dependent on the physical and chemical characteristics of their components, such as the three-dimensional framework of the polymer network, the hydrophilic nature and solubility parameter of the network [48], and the free volume entrapped within the network structure [49,50]. Light curing of the resin cement has shown to increase the monomer conversion of the material and has therefore been found convenient to increase its mechanical properties, to enhance the adhesion to tooth structure, and to decrease the hydrolytic criticalities related to bonding procedures [41,42]. However, plasticizing effects with swelling and fillers disruptions deriving from water entrapment [1,17] can degrade unpolymerized monomers during the curing process, possibly exposing the cement to discoloration risks [51].

It has been previously suggested that the presence of benzoyl peroxide and aromatic tertiary amine necessary to start the polymerization reaction may cause resin cement discoloration effects over time [12,52,53]. Notwithstanding the high yellowish (b* axis) at baseline in dual-cure amine-based resin cement, this effect decreased after aging [36] inducing the authors to suppose an incomplete polymerization with subsequent leaching of unreacted monomers in the oral environment [54,55,56]. Due to these reported backgrounds, research has moved to investigate alternative redox initiator systems. Amine-reduced and the elimination of benzoyl peroxide resulted in minor color changes after aging [53].

In the present study, five one-step self-adhesive resin cements and two universal resin cements were evaluated for their color stability after water aging. Because a standardization of the shade of the cements of different manufacturer’s does not exist, for the purpose of this study the most transparent shades available were chosen for each material to decrease the variabilities related to color differences [30]. Moreover, lighter composite shades are likely to exhibit higher color change even after a short period of water storage [36,54,55,56]. This is consistent with other studies using the lightest shades to evaluate color stability after artificial aging [46,57,58,59].

The spectrophotometric analysis revealed a wide range of discoloration among the tested materials (Table 2) and all the investigated cements have shown color variability below the threshold for clinical acceptability (AT: ΔE* ab = 2.66) and perceptibility (PT: ΔE* ab = 1.22) [60,61,62]. Among the tested materials, MCU showed the lowest color change whereas PSA and RXU showed the highest. This finding is in line with Shiozawa et al. [63], who reported the highest color stability for two shades of Maxcem Elite in respect to all the other investigated cements (Clearfil SA Cement Automix, RelyX Unicem 2 Automix, Super-Bond C&B), after 1 week of water storage.

The optical parameters can be influenced by several factors, such as the amount and type of filler content, the resin-matrix volume, and the matrix-filler bonding. Regarding filler content, it has been seen that a higher percentage of particles is related to lower possibility of water sorption and solubility [64,65]. In other words, the greater the number of particles forming the polymer matrix, the lower the possibility for water to creep into the intraparticle spaces and, consequently, the lower the degree of hydrolytic degradation. Because the degree of water sorption influences the biocompatibility of the material and its mechanical properties, it appears rational that the color stability of the cement will also be favored by a lower degree of water sorption [22].

The size of filler particles also influences the color stability of a composite material, included resin cements, with smaller filler sizes achieving the best results in terms of color stability. The reason for this is the formation of a more homogeneous structure when present with smaller particles, when compared to the polymeric matrix achieved by larger particles (>50 μm) [23,66].

Regarding the matrix composition, urethane dimethacrylate (UDMA) resin-based materials have been reported to be less susceptible to water sorption and therefore more resistant to color change than Bis-GMA-containing cements [43,67].

This could tentatively explain the results obtained by MCU, a UDMA-containing self-adhesive resin cement. Conversely, Panavia SA contains a mixture of TEGDMA, BisGMA and HEMA. BisGMA resins are characterized by high viscosity that is generally attributed to the presence of two hydroxyl groups that are able to establish strong hydrogen bonding interactions [66]. In order to enhance the flowability of the material, additional monomers have been mixed with BisGMA-based cements, such as TEGDMA and HEMA. However, this chemical mix resulted in even higher water uptake which increased swelling of the resin matrix and therefore influenced the color stability of the material. Even though RXU possesses a different chemical composition than PSA, it comparably attained the lowest color stability. This finding was supported by Aldhafyan et al. [68], who reported a similar behavior of these two cements in terms of water sorption and solubility after 30 days of water storage at 37 °C. MCU and RXU belong to a new class of luting material defined as “universal” and characterized by a novel chemical structure that confers a purported versatility which makes it usable in most clinical situations and performs well with most restorative materials, allowing for both adhesive and self-adhesive cementation. The manufacturer (3M ESPE) has developed an amphiphilic redox initiator system and a new filler architecture that is claimed to optimize the rheology characteristics of the material. The new amphiphilic initiator is supposed to form a firm cross-linking polymer network which enables the conversion rate within the hydrophilic cement [33]. However, to the best our knowledge, no data are present on the color stability of this cement after water ageing. Concerning PAS, CUN, and MUL, the inclusion of HEMA in the chemical composition of cements has been reported as detrimental in terms of water sorption, and plasticization phenomena have been observed in previous studies [16,41].

Within the limitation of the present in vitro study, the one-step self-adhesive cements or the latest-introduced universal resin cements used in the self-curing mode demonstrated color changes after water aging. However, these should be considered as preliminary data, as ulterior factors such as a different thickness of the cement layer or restoration, as well as the patient’s saliva composition, make the esthetic results of the restorations challenging. Further studies are necessary to validate the obtained result, possibly improving the simulation of the clinical setting.

## Figures and Tables

**Figure 1 polymers-15-00655-f001:**
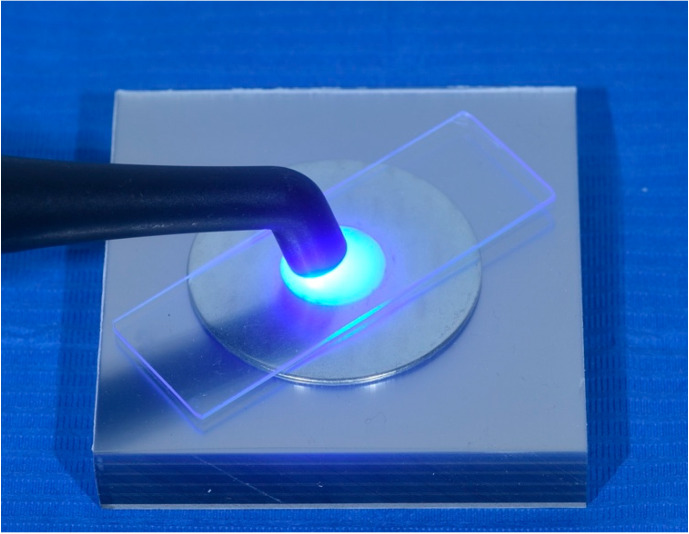
Example of the mold used to prepare the resin cement specimens. Light-curing was performed through the PVC sheet that was used to limit the formation of air-bubbles or structural inhomogeneities within the material.

**Figure 2 polymers-15-00655-f002:**
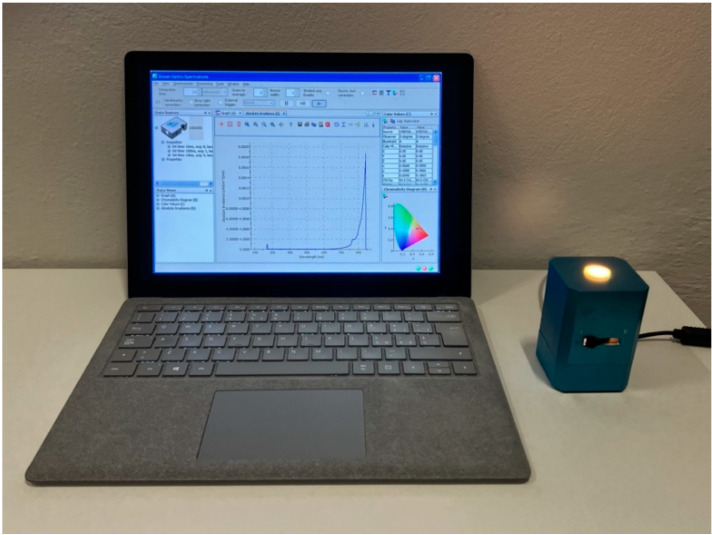
Representation of the spectrophotometer used for the detection of the color of the materials tested in the different laboratory conditions. CIELAB data were collected.

**Figure 3 polymers-15-00655-f003:**
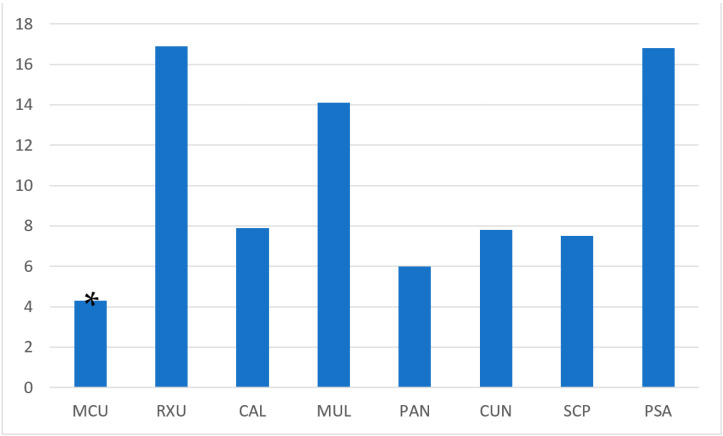
ΔE_ab_ means of the tested resin cements. Differences in color stability were present after water aging, with MXU obtaining the least color variations among groups (asterisk).

**Table 1 polymers-15-00655-t001:** Information of the resin cements investigated according to manufacturers.

Material	Shade	Adhesion Mode	Resin
Maxcem Elite Universal Resin cement MCU (Kerr Corp., Orange, CA, USA)	Clear	Universal	Base: UDMA—Catalyst: Bis-GMA, glycerol dimethacrylate, GPDM. Base: Fluoroalminosilicate glass—Catalyst: Barium aluminoborosilicate glass. Average particle size: 3.5 µm.
RelyX Universal RXU (3M, Seefeld, Germany)	Translucent	Universal	Base: phosphorus oxide, silane, terimethoxyctyl-,hydrolysis product with silica, t-Amyl hydroxiperoxide, n2,6-di-tert-butyl-p-cresol, 2-HEMA, methyl methacylate, acetic acid, copper salt, monohydrate. Catalyst: diurethanedimethacrylate, ytterbium fluoride, glass powder, surface modified with 2-propenoic acid, 2 methyl-3-(trimethoxysilyl)propyl ester and phenyltrimethoxy silane, TEGDMA, L-ascorbic acid, 6-hexadecanoate, hydrate, silane, trimethoxyoctyl, hydrolisis product with silica, 2-HEMA, titanium dioxide, triphenyl phosphite.
Calibra Ceram CAL (Dentsply Sirona, Bernsheim Germany)	Translucent	Self-adhesive	UDMA, di- and tri-methacrylate resins, acrylic resin modified with phosphoric acid, bariumboron- fluoroaluminosilicate glass, organic peroxide as initiator, camphorquinone –(CQ)-photo initiator, phosphine oxide photo initiator, accelerator, butylhydroxytoluene, UV stabilizer, titanium dioxide, iron oxide, hydrophobic amorphous silica. Particle size of the inorganic filler: 16 nm–7 µm; average particle size: 3.8 µm; filler content: 48.7 vol%.
Multilink N MUL (Ivoclar, Schaan, Liechtenstein)	Transparent	Self-adhesive	Multilink Primer A: aqueous solution of initiators Multilink Primer B: HEMA, phosphonic acid monomer, methacrylate monomers cement: dimethacrylate, HEMA, barium glass, ytterbium trifluoride, spheroid mixed oxide (particle size: 0.25–3.0 m; mean filler size: 0.9 m; 40 vol%).
Panavia V5 PAN (Kuraray Noritake, Tokyo, Japan)	Clear	Self-adhesive	Bis-GMA, TEGDMA, new chemical polymerization accelerator, dl-camphorquinone, silanated barium glass filler, silanated aluminum oxide filler, silanated fluoroalminosilicate glass filler (0.01–12 μm average particle size).
Calibra Universal CUN (Dentsply Sirona, Bernsheim Germany)	Translucent	Self-adhesive	Bonding agent: PENTA, 10-MDP, multifunctional acrylate, bifunctional acrylate, camphorquinone /tertiary amine, isopropanol (10–24.5 %), water (5− 24.5 %).—Paste: HEMA, GDM, UDMA, 1,1,3,3-tetramethylbutyl hydroperoxide, TEGDMA, fluoroaluminosilicate glass, GPDM, barium glass filler, fumed silica (69 wt %).
Speed Cem Plus SCP (Ivoclar, Schaan, Liechtenstein)	Transparent	Self-adhesive	UDMA, TEGDMA, PEGDMA, phosphoric acid ester, dibenzoyl peroxide, ytterbium trifluoride, barium glass, silicon dioxide (0.1–7 μm average particle size).
Panavia SA PSA (Kuraray Noritake, Tokyo, Japan)	Translucent	Self-adhesive	Bis-GMA, TEGDMA, MDP, HEMA, dl-camphorquinone, silanated barium glass filler, surface treated sodium fluoride (0.02–20 μm average particle size).

UDMA: urethane dimethacrylate; Bis-GMA: bisphenol A-glycidyl methacrylate; HEMA: 2-hydroxyethyl methacrylate; TEGDMA: triethylene glycol dimethacrylate; PENTA: dipentaerythritol penttacrylate monophosphate; 10_MDP: 10-methacryloxydecyl dihydrogen phosphate; GDM: glycerol 1,3-dimethacrylate; GPDM: glycerol phosphate dimethacryalte; PEGDMA: polyethylene glycol dimethacrylate.

**Table 2 polymers-15-00655-t002:** Descriptive statistics of the recorded color differences values in ΔE_ab_ (mean ± standard deviation); in the significance column different letters label statistically significant between-group differences.

		Color Difference
Material	ΔE_ab_	Significance *p* < 0.05
MCU	4.3 ± 0.7	a
RXU	16.9 ± 1.6	e
CAL	7.9 ± 1	c
MUL	14.1 ± 0.7	d
PAN	6 ± 0.7	b
CUN	7.8 ± 1.3	c
SCP	7.5 ± 0.8	c
PSA	16.8 ± 1.2	e

**Table 3 polymers-15-00655-t003:** Absolute values of the color parameters L*, a* and b* recorded before and after water aging, and calculated ΔL*, Δa*, Δb*, ΔEab.

Materials	Before Aging			After Aging							
MAX	L*	a*	b*	L*	a*	b*		ΔL*	Δa*	Δb*	ΔE_ab_
	75.40	−1.10	4.20	80.80	−1.50	3.00		−5.40	0.40	1.20	5.5
	76.30	−0.80	4.40	80.40	−1.30	4.10		−4.10	0.50	0.30	4.1
	76.90	−0.80	4.40	79.70	−1.00	2.80		−2.80	0.20	1.60	3.2
	76.10	−0.80	4.30	80.10	−1.10	2.80		−4.00	0.30	1.50	4.3
	76.40	−0.80	4.70	80.30	−1.30	3.80		−3.90	0.50	0.90	4.0
	75.80	−0.80	4.90	79.20	−1.30	3.30		−3.40	0.50	1.60	3.8
	74.40	−0.90	5.00	79.30	−1.40	4.20		−4.90	0.50	0.80	5.0
	74.80	−0.60	4.30	79.90	−1.20	4.00		−5.10	0.60	0.30	5.1
	76.80	−0.70	4.10	80.20	−1.30	3.60		−3.40	0.60	0.50	3.5
	76.50	−0.70	4.10	80.70	−1.20	3.40		−4.20	0.50	0.70	4.3
							Mean	−4.12	0.46	0.94	4.3
RXU	89.10	−2.20	9.20	89.10	−4.40	23.90		0.00	2.20	−14.70	14.9
	90.30	−2.50	10.60	88.00	−4.70	31.00		2.30	2.20	−20.40	20.6
	89.60	−2.30	12.10	88.50	−4.00	27.00		1.10	1.70	−14.90	15.0
	88.40	−1.90	10.30	87.60	−3.30	27.30		0.80	1.40	−17.00	17.1
	89.30	−2.20	10.50	89.10	−4.20	27.70		0.20	2.00	−17.20	17.3
	89.50	−1.60	10.00	88.70	−2.70	27.60		0.80	1.10	−17.60	17.7
	88.40	−3.30	11.50	86.50	−4.80	28.10		1.90	1.50	−16.60	16.8
	88.90	−2.70	11.90	87.30	−4.60	28.10		1.60	1.90	−16.20	16.4
	89.10	−2.50	11.10	88.50	−4.40	27.10		0.60	1.90	−16.00	16.1
	89.00	−2.00	12.10	88.70	−4.00	28.70		0.30	2.00	−16.60	16.7
							Mean	0.96	1.79	−16.72	16.9
CAL	55.70	4.60	23.80	60.50	2.20	18.50		−4.80	2.40	5.30	7.5
	55.90	4.30	23.10	60.20	2.00	17.70		−4.30	2.30	5.40	7.3
	55.00	4.40	23.40	62.30	1.70	18.30		−7.30	2.70	5.10	9.3
	57.30	4.20	23.40	63.50	1.80	17.30		−6.20	2.40	6.10	9.0
	55.60	4.30	23.30	61.10	1.40	17.80		−5.50	2.90	5.50	8.3
	55.70	4.60	23.90	61.30	2.00	20.40		−5.60	2.60	3.50	7.1
	55.00	4.60	24.00	60.60	1.30	18.60		−5.60	3.30	5.40	8.5
	55.70	4.80	23.30	60.60	2.30	20.50		−4.90	2.50	2.80	6.2
	55.50	4.30	22.10	60.40	2.00	17.90		−4.90	2.30	4.20	6.9
	55.40	4.70	23.70	60.40	2.20	17.10		−5.00	2.50	6.60	8.6
							Mean	−5.41	2.59	4.99	7.9
MUL	77.00	−2.60	21.80	67.70	5.10	29.20		9.30	−7.70	−7.40	14.2
	78.30	−3.20	20.70	67.00	3.50	25.70		11.30	−6.70	−5.00	14.1
	77.80	−2.80	22.00	69.00	4.60	29.20		8.80	−7.40	−7.20	13.6
	76.00	−2.10	20.40	66.90	4.40	26.60		9.10	−6.50	−6.20	12.8
	76.20	−2.50	20.90	66.10	4.70	27.40		10.10	−7.20	−6.50	14.0
	76.80	−3.10	20.40	66.90	4.80	26.90		9.90	−7.90	−6.50	14.2
	76.10	−3.20	20.70	66.00	4.60	26.40		10.10	−7.80	−5.70	14.0
	75.60	−3.00	20.60	65.20	5.20	29.00		10.40	−8.20	−8.40	15.7
	76.30	−3.30	21.40	66.90	4.90	29.30		9.40	−8.20	−7.90	14.8
	74.60	−3.30	20.10	65.60	4.70	27.30		9.00	−8.00	−7.20	14.0
							Mean	9.74	−7.56	−6.80	14.1
PAN	72.50	−1.40	1.00	75.40	−3.10	5.70		−2.90	1.70	−4.70	5.8
		−1.50	1.50	76.90	−3.20	6.50		−4.90	1.70	−5.00	7.2
	72.30	−1.40	1.00	75.00	−2.80	5.10		−2.70	1.40	−4.10	5.1
	71.90	−1.50	1.10	75.20	−3.20	6.30		−3.30	1.70	−5.20	6.4
	73.20	−1.50	1.50	76.90	−3.60	7.00		−3.70	2.10	−5.50	7.0
	71.70	−1.40	1.50	74.50	−3.20	5.40		−2.80	1.80	−3.90	5.1
	73.30	−0.70	0.80	76.10	−3.60	4.70		−2.80	2.90	−3.90	5.6
	71.60	−1.00	1.60	74.90	−3.80	6.00		−3.30	2.80	−4.40	6.2
	71.60	−1.70	1.80	74.80	−3.60	6.30		−3.20	1.90	−4.50	5.8
	71.40	−1.40	1.60	74.70	−3.50	6.20		−3.30	2.10	−4.60	6.0
							Mean	−3.29	2.01	−4.58	6.0
CUN	42.50	5.80	21.70	47.20	3.50	18.40		−4.70	2.30	3.30	6.2
	47.00	6.60	27.60	53.20	4.30	21.60		−6.20	2.30	6.00	8.9
	47.70	6.30	25.10	54.40	4.90	20.40		−6.70	1.40	4.70	8.3
	46.70	5.70	21.60	54.50	3.80	18.60		−7.80	1.90	3.00	8.6
	45.50	5.80	23.50	51.40	4.20	20.10		−5.90	1.60	3.40	7.0
	45.00	5.40	22.70	51.60	3.80	19.10		−6.60	1.60	3.60	7.7
	41.00	5.80	21.00	46.40	4.30	17.50		−5.40	1.50	3.50	6.6
	43.50	5.80	23.70	52.00	3.90	21.00		−8.50	1.90	2.70	9.1
	43.40	5.20	22.50	48.80	3.80	19.70		−5.40	1.40	2.80	6.2
	44.70	5.50	24.50	52.40	3.00	19.20		−7.70	2.50	5.30	9.7
							Mean	−6.49	1.84	3.83	7.8
SCP	88.50	0.00	12.90	90.90	−1.70	19.20		−2.40	1.70	−6.30	7.0
	88.00	−0.30	12.90	90.20	−2.20	21.00		−2.20	1.90	−8.10	8.6
	88.80	−0.20	11.90	89.90	−1.40	18.30		−1.10	1.20	−6.40	6.6
	88.60	−0.60	12.80	90.10	−1.90	20.10		−1.50	1.30	−7.30	7.6
	88.20	−0.60	12.00	89.80	−1.90	19.10		−1.60	1.30	−7.10	7.4
	88.60	−0.80	11.90	89.60	−2.30	19.40		−1.00	1.50	−7.50	7.7
	85.80	−0.60	12.60	88.10	−2.10	20.90		−2.30	1.50	−8.30	8.7
	85.80	−0.30	12.90	87.00	−1.90	19.00		−1.20	1.60	−6.10	6.4
	86.10	−0.60	12.10	87.90	−2.00	19.90		−1.80	1.40	−7.80	8.1
	86.90	−0.50	12.30	88.40	−1.90	18.90		−1.50	1.40	−6.60	6.9
							Mean	−1.66	1.48	−7.15	7.5
PSA	84.40	−5.00	9.70	82.90	−4.90	27.80		1.50	−0.10	−18.10	18.2
	85.60	−4.90	10.20	81.70	−3.00	27.80		3.90	−1.90	−17.60	18.1
	85.20	−5.10	10.60	85.10	−4.90	25.10		0.10	−0.20	−14.50	14.5
	85.20	−4.90	10.50	83.10	−3.10	26.10		2.10	−1.80	−15.60	15.8
	85.20	−4.80	9.80	82.40	−3.80	27.60		2.80	−1.00	−17.80	18.0
	85.20	−4.90	10.40	82.90	−4.00	26.80		2.30	−0.90	−16.40	16.6
	85.30	−5.10	9.80	83.70	−4.00	25.80		1.60	−1.10	−16.00	16.1
	84.80	−4.80	10.30	82.90	−3.70	26.50		1.90	−1.10	−16.20	16.3
	85.00	−4.80	9.50	83.10	−4.30	26.00		1.90	−0.50	−16.50	16.6
	84.90	−5.00	10.10	81.40	−4.40	26.90		3.50	−0.60	−16.80	17.2
							Mean	2.16	−0.92	−16.55	16.8

## Data Availability

The data presented in this study are available on request from the corresponding author. The data are not publicly available due to the university’s policy on access.

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
