# Peer review of "Color Stability of Resin Cements after Water Aging"

_polymers, 2023, doi:10.3390/polym15030655_

Round 1
Reviewer 1 Report
Well designed research. The methods are adequately described and the results - clearly presented.
Question 1
You correctly mention, that a standardization of the shade of the cements of different manufacturer's does not exist and you have chosen the most transparent ones. But from table 1 is evident that some of the cements are translucent. From scientific point of view it would be more correct the starting point to be one and the same. It doesn't interfere with your results, since you use spectrophotometry, which is a very reliable method, but some kind of calibration is advocated.
My second comment is about the method of aging you have chosen. In my opinion - THERMOCYCLING is a better option. You store the samples in a basket dipped in 20degr. fluid (distilled water) for 30sec., then adjust the apparatus to place the samples into a tank filled with 50 degr. fluid for another 30 sec. and this can be repeated as many times as you set the device. Although there isn't an agreement about the number of cycles that are relevant to an year stay in the mouth for example, some 5000 - 6000 cycles would give quite representative results. But this is only a personal recommendation, of course, based on my practical experience.
Author Response
Well designed research. The methods are adequately described and the results - clearly presented.
Question 1
You correctly mention, that a standardization of the shade of the cements of different manufacturer's does not exist and you have chosen the most transparent ones. But from table 1 is evident that some of the cements are translucent. From scientific point of view, it would be more correct the starting point to be one and the same. It doesn't interfere with your results, since you use spectrophotometry, which is a very reliable method, but some kind of calibration is advocated.
Our response: We thank the reviewer for taking the time to read the article and for his/her comments which allow us to clarify a very important aspect which is that of the color of resinous cements. The authors agree with the reviewer that, from a scientific point of view, it would be appropriate to have materials of the same shade. Unfortunately, this is practically (and clinically) almost never possible when products from different manufacturers are tested (Carrabba M, Vichi A, Tozzi G, Louca C, Ferrari M. Cement opacity and color as influencing factors on the final shade of metal-free ceramic restorations. J Esthet Rest Dent 022 Mar;34(2):423-429; Tabatabian F, Bakhshaei D, Namdari M. Effect of Resin Cement Brand on the Color of Zirconia-Based Restorations. J Esthet Rest Dent 2020 Apr;29(4):350-355). Recent literature studies have identified that, even if classified with the same shade, products from different manufacturers do not correspond to each other (Tabatabian F, Bakhshaei D, Namdari M. Effect of Resin Cement Brand on the Color of Zirconia-Based Restorations. J Esthet Rest Dent 2020 Apr;29(4):350-355; Gomes C, Martins F, Reis JA, Albacete-Martinez CP, Maurício PD. Final esthetic result of ceramic restorations cemented with different colors of cement. Clin Exp Dent Res 2022 Feb;8(1):257-261). For this reason, the decision to select the clearest shapes declared for each product was considered as the most reasonable one. The spectrophotometric evaluation at the baseline allows to draw up a starting point for each material and to evaluate the color change after the staining or aging procedures.
My second comment is about the method of aging you have chosen. In my opinion – THERMOCYCLING is a better option. You store the samples in a basket dipped in 20degr. Fluid (distilled water) for 30sec., then adjust the apparatus to place the samples into a tank filled with 50 degr. Fluid for another 30 sec. and this can be repeated as many times as you set the device. Although there isn’t an agreement about the number of cycles that are relevant to an year stay in the mouth for example, some 5000 – 6000 cycles would give quite representative results. But this is only a personal recommendation, of course, based on my practical experience.
Our response: We would like to thank the reviewer to share with us his/her opinion, that we respect. However, according to ISO standards (7491-2000 and 4049-2019 7.13) and literature information, aging methods can be mainly divided into two categories: “static” and “dynamic”. The static one refers to immersion in water as aging procedure, that is the method used in our study, while the dynamic one foresee procedures such as themocycling. However, according to the information cited before, there is no conclusive evidence whether one method could be better than the other, even if in both the cited standards a static method is indicated. As previously reported in literature of our paper, (Asmussen, E. An Accelerated Test for Color Stability of Restorative Resins. Acta Odontologica Scandinavica 1981, 39, 329–332, doi:10.3109/00016358109162704; Vichi, A.; Ferrari, M.; Davidson, C.L. Color and Opacity Variations in Three Different Resin-Based Composite Products after Water Aging. Dent. Mater. 2004, 20, 530–534, doi:10.1016/j.dental.2002.11.001), the “static” water aging is a reliable procedure to be used to evaluate the influence on colour stability of resinous materials. Indeed, when dealing with colour evaluation, it seems that material’s discoloration is more related to water hydrolysis (Espíndola-Castro, L.F.; Brito, O.F.F. de; Araújo, L.G.A.; Santos, I.L.A.; Monteiro, G.Q.D.M. In Vitro Evaluation of Physical and Mechanical Properties of Light-Curing Resin Cement: A Comparative Study. Eur. J. Dent. 2020, 14, 152–156, doi:10.1055/s-0040-1705075; Mina, N.R.; Baba, N.Z.; Al-Harbi, F.A.; Elgezawi, M.F.; Daou, M. The Influence of Simulated Aging on the Color Stability of Composite Resin Cements. J. Prosth. Dent. 2019, 121, 306–310, doi:10.1016/j.prosdent.2018.03.014), more than to the thermal stresses that, from their sides, would instead impact on the mechanical properties of the material itself.
Reviewer 2 Report
This a well planed study and the authors should be congratulated for it.
I do have some suggestions
Lines 38-48 need references
Table 1 is too lengthy making it unreadable due to the columns being of the same size. please shrink some and make the last one wider. Also what is the source of the data
Table 1 legend is confusing
Line 107 The most current ISO 4049:2019
Line 135 I was very happy with this line but the following one 136 had me unsettled. why didn't the authors used what is standard, a grey background? what is the rational for the use of a ceramic bloc as a background? Also what was the bloc thickness? what was the color behind the bloc?
In Table 1 there are codes for each cement, on table 2 there are groups number 1-8, but then on figure 3 we are back to codes. Please only one type of codification/grouping
Lines 203-220 have no place in the discussion and are not relevant to the study.
Lines 221 -238 are the authors trying to explain their results or are this just random facts ? did the authors make mechanical tests?
Line 242 replace swallowing with swelling
Lines 261-267 did the authors understand the results:
even though none of the investigated cements have shown a discoloration below the threshold for clinical acceptability (AT: ΔE* ab = 2.66) and perceptibility (PT: ΔE* ab = 1.22) ALL DID MCU is the lowest with 4.3
Lines 269-281 are a short review of the science but without a clear explanation on what is relevant to the manuscript:
The optical parameters can be influenced by several factors, such as the amount and type of filler content, the resin matrix volume, and the matrix-filler bonding. what fillers? which ones?
The size of filler particles also influences the color stability what is the size?
Also the previous statement is it for composite or cements?
Lines 308-313 are very speculative since the authors tested the materials in non-clinical ways/settings/indications. This type of advice based on a high failure test setting is unbecoming.
The funding/Acknowledgments are contradicting. please explain
Is there a need to self cite 6 studies published in 2022?
Author Response
This a well planed study and the authors should be congratulated for it.
I do have some suggestions
Lines 38-48 need references.
Our response: Thanks to the reviewer for taking the time to read the manuscript. We appreciate the comments you provide which we are sure will lead to improvements to the article. According to the suggestion, references have been added to the text.
Table 1 is too lengthy making it unreadable due to the columns being of the same size. Please shrink some and make the last one wider. Also what is the source of the data.
Our response: Thank you for pointing out this problem. Accordingly, the table was re-arranged.
Table 1 legend is confusing
Our response: The legend of Table 1 was reformulated to avoid confusions to the reader.
Line 107 The most current ISO 4049:2019
Our response: Thank you for the comment. This was mainly a typo of the text, as this ISO standard (4049:2019) was the one present in the reference. This was corrected in the Materials and Methods section, as suggested.
Line 135 I was very happy with this line but the following one 136 had me unsettled. Why didn’t the authors used what is standard, a grey background? What is the rational for the use of a ceramic bloc as a background? Also what was the bloc thickness? What was the color behind the bloc?
Our response: The authors agree with the reviewer that conventionally a 50% grey background is used for color measurement, and they also used this background in several previously published papers. However, in the absence of a specific ISO standard, the use of a different backgrounds is not precluded. In the specific case, due to the translucent cements’ shades selected, the influences of the 50% grey background was considered by the authors to excessively lower the L* Value, far away from the clinical condition. For this reason, a ceramic MARK II A3 I-40 CAD-CAM block, that was considered more similar to an average dentin color, was instead used as background, The I40 block used has the dimension of 15,5 x 19 x 39 mm, therefore the measurement was performed using as background the largest surface available, that is 39 x 19 mm (opening of the integrating sphere is 10 mm) with a thickness of 15,5 mm. As conventionally the absolute color, that is the thickness after which the background does not influence anymore the color of dental composites/ceramics, is 5 mm, the block used largely exceeded this dimension (> 10 mm), thus no further background was placed on the block.
The information was added to the Materials and Methods section.
In Table 1 there are codes for each cement, on table 2 there are groups number 1-8, but then on figure 3 we are back to codes. Please only one type of codification/grouping
Our response: Thank you to the Reviewer for the suggestion. All the tables were checked out and codes were adjusted.
Lines 203-220 have no place in the discussion and are not relevant to the study.
Our response: Thank you for the comment. This part was reformatted and reduced according to the Reviewer’s suggestions.
Lines 221 -238 are the authors trying to explain their results or are this just random facts ? did the authors make mechanical tests?
Our response: Thank you for the comment. This study aimed at evaluating the color stability of different resin cements when submitted to water aging, and no mechanical tests were considered. In order to avoid misleading to the reader, this part has been slightly modified as highlighted in the text.
Line 242 replace swallowing with swelling
Our response: This was done and highlighted in the text.
Lines 261-267 did the authors understand the results:
even though none of the investigated cements have shown a discoloration below the threshold for clinical acceptability (AT: ΔE* ab = 2.66) and perceptibility (PT: ΔE* ab = 1.22) ALL DID MCU is the lowest with 4.3.
Our response: Thanks to the Reviewer for the comment. We agree that in the present form, the sentence could be misleading. Accordingly, this was modified and properly highlighted through the text.
Lines 269-281 are a short review of the science but without a clear explanation on what is relevant to the manuscript:
Our response: Thanks the reviewer for the comment. Color changes in resinous materials (both composite and cements) have been indicated to be related to intrinsic factors (material’s composition such has filler composition and dimension, type of photoinitiators ecc) or extrinsic factors (this is the case of absorption of external dyes such as coffee or other staining solutions). Since the present study aimed to evaluate the effects of water aging on the color stability of different resin cements, considerations of the intrinsic factors of the materials were made (i.e. filler particles) and the Authors feel that the effects of this parameter, based on literature information, was mandatory to understand the results obtained in the study.
The optical parameters can be influenced by several factors, such as the amount and type of filler content, the resin matrix volume, and the matrix-filler bonding. what fillers? which ones?
The size of filler particles also influences the color stability what is the size?
Also the previous statement is it for composite or cements?
Our response: Thank you for the comment. Resinous dental materials (both composite and cements) present a similar structure formed by a matrix and filler contents. the filler component is mainly responsible for the color of the material (Suh YR, Ahn JS, Ju SW, Kim KM. Influences of filler content and size on the color adjustment potential of nonlayered resin composites. Dent Mater J 2017; 31:35–40). Differences in the filler grain sizes and contents allow a classification into macro-, micro- or nano-filled materials (Ferracane JL. Resin composite—state of the art. Dent Mater 2011;27:29–38.). Several studies have focused on the effect of filler type, shape, and size on the resin-based composites’ overall appearance as well as their ageing resistance (J.J. Kim, H.J. Moon, B.S. Lim, Y.K. Lee, S.H. Rhee, H.C. Yang, The effect of nanofiller on the opacity of experimental composites, J. Biomed. Mater. Res. B Appl. Biomater. 80B (2007) 332–338; V.E. Salgado, L.M. Cavalcante, N. Silikas, L.F. Schneider, The influence of nanoscale inorganic content over optical and surface properties of model composites, J. Dent. 41 (5) (2013) e45–53; V.E. Salgado, P.P. Albuquerque, L.M. Cavalcante, C.S. Pfeifer, R.R. Moraes, L.F. Schneider, Influence of photoinitiator system and nanofiller size on the optical properties and cure efficiency of model composites, Dent. Mater. 30 (10) (2014) e264–71). Resin cement with a higher percentage of filler content has lower water sorption and solubility (Alshali RZ, Salim NA, Satterthwaite JD, Silikas N. Long- term sorption and solubility of bulk-fill and conventional resin-composites in water and artificial saliva. J Dent 2015;43(12):1511–1518). Sorption and solubility can influence the biocompatibility, mechanical properties, and color stability of resin cements (Marghalani HY. Sorption and solubility characteristics of self-adhesive resin cements. Dent Mater 2012;28(10): e187–e198). The staining of resin cement may be caused by intrinsic (filler content, material composition, or type of activation). Though the size of fillers in dental resin-based composites has been the subject of many articles, resin materials composed with smaller fillers showed improved color stability and gloss retention (Salgado VE, Cavalcante LM, Moraes RR, Davis HB, Ferracane JL, Schneider LF. Degradation of optical and surface properties of resin-based composites with distinct nanoparticle sizes but equivalent surface area. J Dent 2017;59(1):48–53).
In accordance, information was added to this part of the discussion.
Lines 308-313 are very speculative since the authors tested the materials in non-clinical ways/settings/indications. This type of advice based on a high failure test setting is unbecoming.
Our response: We thank the Reviewer for the comment. The Authors agree with her/him that this part could be misleading and was therefore modified.
The funding/Acknowledgments are contradicting. please explain
Our response: This section was adjusted to avoid misleading.
Is there a need to self cite 6 studies published in 2022?
Our response: We thank the Reviewer for the comment. Considering that the manuscript was written by a group of researchers who have been working on a series of continued research related to the world of color evaluation and resin cements in restorative and prosthetic dentistry and have significantly contributed to this topic, we feel that self-citing, in this particular case, is not considered as bad practice (please, see the reference: Hemmat Esfe M, Wongwises S, Asadi A, Karimipour A, Akbari M. Mandatory and self-citation; types, reasons, their benefits and disadvantages. Science and Engineering Ethics. 2015 Dec;21(6):1581-5), additionally considering that 6 citations out of 66 represent less than 3% of the total references.
Round 2
Reviewer 2 Report
Thank you for accepting my comments and for your explanations
Author Response
Thank you to the Reviewer for the time spent to review this article.